# Ozone Efficacy for the Disinfection of Ambulances Used to Transport Patients during the COVID-19 Pandemic in Peru

**DOI:** 10.3390/ijerph20105776

**Published:** 2023-05-10

**Authors:** Miguel Alejandro Gómez-Castillo, Cristina Rivera Romero, Kevin Reátegui-Ochoa, Enrique Mamani Zapana, Marcial Silva-Jaimes

**Affiliations:** 1Laboratorio de Microbiología de Alimentos, Facultad de Industrias Alimentarias, Universidad Nacional Agraria La Molina, Lima 15024, Peru; mgomez@lamolina.edu.pe (M.A.G.-C.); kevin.reate.ochoa@gmail.com (K.R.-O.); 2Facultad de Zootecnia, Universidad Nacional Agraria La Molina, Lima 15024, Peru; crivera@lamolina.edu.pe; 3Facultad de Biología, Universidad Nacional Mayor de San Marcos, Lima 15081, Peru; emamaniz@unmsm.edu.pe

**Keywords:** disinfection, ozone, SARS-CoV-2, RT-qPCR, ambulances, surface sampling, hospital pathogens

## Abstract

We assessed the disinfection efficacy of an ozone generator prototype in ambulances used to transport patients with coronavirus disease (COVID-19). This research consisted of three stages: in vitro tests using microbial indicators, such as *Candida albicans, Escherichia coli, Staphylococcus aureus* and *Salmonella phage*, which were experimentally inoculated onto polystyrene crystal surfaces within a 23 m^3^ enclosure. They were then exposed to ozone at a 25 ppm concentration using the ozone generator (Tecnofood SAC) portable prototype, and the decimal reduction time (D) was estimated for each indicator. The second stage involved the experimental inoculation of the same microbial indicators on a variety of surfaces inside conventional ambulances. The third stage consisted of exploratory field testing in ambulances used to transport patients with suspected COVID-19. During the second and third stages, samples were collected by swabbing different surfaces before and after 25 ppm ozonisation for 30 min. Results suggested that ozone was most effective on *Candida albicans* (D = 2.65 min), followed by *Escherichia coli* (D = 3.14 min), *Salmonella phage* (D = 5.01 min) and *Staphylococcus aureus* (D = 5.40 min). Up to 5% of the microbes survived following ozonisation of conventional ambulances. Of the 126 surface samples collected from ambulances transporting patients with COVID-19, 7 were positive (5.6%) for SARS-related coronavirus as determined on reverse transcription quantitative real-time polymerase chain reaction (RT-qPCR). Ozone exposure from the ozone generator prototype inside ambulances at a concentration of 25 ppm for 30 min can eliminate gram positive and negative bacteria, yeasts, and viruses.

## 1. Introduction

At the beginning of the pandemic in Peru, the transmission capacity of SARS-CoV-2, the causative agent of COVID-19, was quite high. Chin et al. [1] and Van Doremalen et al. [2] suggested that SARS-CoV-2 remains viable from 3 h to 7 days, depending on the type of surface. COVID-19 is a disease that probably originated from the increased interaction among wild animals, farm animals and human beings [3]. It has been classified as a pandemic as it is caused by a new influenza-type virus, which has spread worldwide, and most people are not immune to it. Currently, statistics in Peru show 16.3% positivity with a total of 1,734,606 positive cases since the first outbreak of the disease in the country. Hospital settings, ambulances and gurneys used to transport and manage patients with COVID-19 are high risk areas for disease transmission. Hence, the risk of virus transmission should be reduced using fast and reliable methods to disinfect those areas that have been exposed to patients with suspected or confirmed SARS-CoV-2 infection. Ozone has been established as an agent for the disinfection and elimination of bacteria, mould, viruses, and toxins from the surfaces of equipment and products. Some of its applications are associated with the disinfection of food, water, and food management areas; it has also been used in medicine and in the treatment of hospital effluents to avoid secondary virus transmission [4,5]. Furthermore, some studies have verified the efficacy of ozone against SARS [6,7,8]. For this reason, the ozonisation of the interior surfaces of ambulances may contribute to the elimination of bacteria, mould, and viruses (such as SARS), thus protecting medical staff against potential infections, which has been observed during this pandemic. The purpose of this research is to assess the efficacy of the ozone generator (OzGe) prototype, which can produce 25 ppm ozone, for the disinfection of the interior surfaces of ambulances.

## 2. Materials and Methods

### 2.1. Type of Study

An experimental analytical study was conducted in order to assess the effect of ozone exposure on the disinfection of the interior surfaces of ambulances. The study consisted of three stages: in vitro tests, field testing with experimental inoculation inside conventional ambulances, and exploratory testing in ambulances transporting patients with COVID-19.

### 2.2. Ozone Generation Prototype

The OzGe prototype has been manufactured by Tecnofoods SAC, Lima, Peru; its dimensions are 150 × 60 × 60 cm in size, weighs 42 kg, has a 300 m^3^ disinfection coverage and a 1000 m^3^ deodorisation coverage, and generates a maximum ozone concentration of 25 ppm. Inside the prototype are placed 4 UV-C lamps, each one of 89 W and 10,000-h duration; the lamps were 1 m in length and 5 cm in diameter. In the three experimental stages, the temperature during the ozonisation process was 26 ± 2 °C, and the relative humidity (RH) before and after ozonisation was 68–75% and 50–55%, respectively. 

### 2.3. In Vitro Tests

This stage was carried out inside a 2.29 × 3.33 × 3.05 m enclosure within the Food Microbiology Laboratory (in Spanish, LMA) of Universidad Nacional Agraria La Molina (UNALM) facility using OzGe. *Staphylococcus aureus* (SA) (ATCC 25923) and *Escherichia coli* (EC) (ATCC 25922) strains provided by the LMA of UNALM as well as *Candida albicans* (CA) strains (ATCC 10231) and isolated strains of *Salmonella infantis* (SI) and *Salmonella phage* 21 (Φ21) provided by the Virology Laboratory of Universidad Nacional Mayor de San Marcos were used in this study. Suspensions of SA, EC and CA were prepared, which were equivalent to 0.5 McFarland standard. A suspension equivalent to 10^8^ PFU/mL of Φ21 was prepared based on the indications given by Jamalludeen et al. [9]; 500 μL of log-phase SI and 500 μL of Φ21 were added to 40 mL of tryptic soy broth, which was later incubated at 37 °C for 24 h. The preparation was then centrifuged at 8000 rpm for 10 min; the supernatant recovered was filtered (0.45 μm) and the phage titre was determined by following the double agar overlay method described by Adams [10]: serial 1:10 dilutions up to a 10^−8^ dilution of the bacteriophage sample were performed. A volume of 1 mL of each dilution was then mixed with 1 mL of log-phase SI in 15 mL empty sterile conical tubes. After leaving this mixture undisturbed for 10 min, 8 mL of 70% tryptic soy agar was added. The preparation was immediately homogenised and poured onto a Petri dish with a thin agar base layer (approximately 10 mL). This was followed by 24 h incubation at 37 °C. In order to estimate the concentration of each dilution, PFUs (plaque forming units) were calculated using the following formula: (1/dilution) × PFU.

Experimental inoculation was conducted based on the method described by Chin et al. [1]; in a biosafety cabinet, 400 μL of standardised inoculum was added to a single-use, sterile, 90 mm diameter Petri plaque (crystal polystyrene, 90 × 15 mm, SPL Life Science, Pocheon City, South Korea). The inoculum was then spread with a sterile swab moistened in buffered peptone water and left to dry at room temperature for 30 min. After the adhesion of microorganisms, the Petri plaques were moved into a 23 m^3^ experimental enclosure. They were opened and ozonised; the OzGe located in the room was turned on until the ozone concentration reached 25 ppm as measured using the ozone gas sensor Aeroqual^TM^ (Aeroqual Limited, Avondale, New Zealand); it reached 25 ppm after 1 h. Once a 25 ppm concentration was achieved, exposure times of 5, 10, 15, 20, 25 and 30 min were recorded. Following completion of each exposure time, the OzGe was turned off. When the ozone concentration decreased to <0.05 ppm, samples were collected to quantify the concentration of the surviving microorganisms according to the protocols described by Adams [10] and the ICMSF [11]. Decimal reduction time was estimated according to the protocol proposed by Gava et al. [12].

### 2.4. Field Testing with Experimental Inoculation in Conventional Ambulances

This stage was carried out in the courtyard of Tecnofoods SAC using conventional ambulances (n = 3) provided by Ambulancias Golden Cross Peru SAC, Lima, Lima Peru. For this field test, the same strains and OzGe used in the in vitro tests were used. The artificial inoculation protocol followed the method proposed by Lang et al. [13]; 200 μL of CA, EC, SA and Φ21 standardised inoculum was dispersed throughout a 25 cm^2^ area (5 cm × 5 cm) using a swab moistened in buffered peptone water. This procedure was performed in three different locations inside conventional ambulances (seat, gurney and shelves) and was tested in triplicate; these locations were suggested by the health workers of the ambulances because they had a significant contact and possible transmission between the people inside the ambulance. The inoculated surfaces were left to dry at room temperature for 30 min, after which the ozonisation process began: the OzGe was placed inside the ambulance, and the surfaces were exposed to ozone until its concentration reached 25 ppm as measured by the ozone gas sensor Aeroqual^TM^ (Aeroqual Limited). After reaching 25 ppm, a 30 min exposure time was recorded. Then, the equipment was turned off, and the lateral and rear doors of the ambulance were opened to let the gas out. Samples were then collected to quantify surviving microorganisms according to the protocols by Adams [10] and the ICMSF [11]. Upon completion, the conventional disinfection protocol, consisting of a 0.1% sodium hypochlorite solution, was applied inside the ambulances to prevent further contamination. Finally, the survival rate (%SR) was determined using the following formula: (%SR = ((LgN_o_ − LgN_f_)/LgN_o_) × 100%)).

### 2.5. Exploratory Field Testing in Ambulances Transporting Patients with COVID-19

This stage was carried out in the solid waste area of the Hospital Nacional Hipólito Unanue (HNHU). The sampled ambulances had an internal volume of 8–13 m^3^. A total of 126 samples were collected at different spots (gurney, seat, ventilation grid, control panel, oxygen valve, gurney belts, railing, medical instrument handles, medical bag handles and lamps) by swabbing different surfaces (plastic, leather, stainless steel, cloth and aluminium) inside ambulances used to transport patients with SARS-CoV-2 infection (n = 3). Samples were collected from the ambulances before and after ozonisation, which consisted of a 30-min exposure of the interior surfaces to a 25-ppm ozone concentration using the OzGe.

Samples were collected with the CITOSWAB-Collection and Transport Kit^TM^ (CITOTEST, Jiangsu, China); samples were transported at a room temperature of 23 ± 2 °C and were refrigerated at 5 °C in the LMA until their analyses. Viral RNA was extracted using specific lysis with silica gel membrane purification technology (Viral RNA Isolation^TM^, Mancherey Nagel, Düren, Germany). Viral RNA concentration and purity were then measured with a spectrophotometer (DeNovix^TM^, Cat. No. DS-11). Extracted RNA was amplified with a reverse transcription, real-time polymerase chain reaction kit (RT-qPCR), which is a qualitative assay to detect SARS-CoV-2 (Vircell Microbiologists, Granada, Spain). This method was based on the amplification of specific SARS-CoV-2 fragments (Wuham-2019-nCoV) and SARS-related coronaviruses in the same reaction tube with RT-qPCR. The Vircell CoV-2 kit contains retrotranscriptase, Taq polymerase, buffer, oligonucleotides with specific probes for nCoV N-gene and an RNA fragment as internal controls (Mix A), as well as E gene of SARS-related coronaviruses and oligonucleotides/probes with human RNase P as targets (Mix B). The positive and negative controls were a solution made of non-infectious nucleic acids (Vircell CoV-2) and deionised water, respectively.

Microbial death curves for the microbial indicators were studied through linear regression analyses in the in vitro stage; the non-parametric Kruskal–Wallis test was used to compare average survival percentages in the second stage field tests. Data obtained were analysed using Minitab 17^TM^ statistical package and Microsoft Office Excel^TM^ software Version 21th.

## 3. Results

A 25 ppm ozone exposure for 15 min can eliminate CA and EC, whereas a 30 min exposure is needed to eliminate SA and the bacteriophages. Moreover, the R^2^, coefficient of determination, was >90% in all cases. Lastly, the decimal reduction value (D) was estimated to determine resistance to ozonisation; SA showed the highest resistance (D = 5.01), whereas CA showed the lowest resistance (D = 2.65) (Figure 1).

Table 1 shows the %SR of the four microbial indicators inoculated onto the surfaces of the seats, gurney and shelves inside conventional ambulances exposed to a 25 ppm ozone concentration for 30 min. After ozonisation, 1% and 3.1% of the SA population inoculated on the seats and gurney survived, respectively, whereas 5.3% of the inoculated bacteriophages survived on the shelves. The effect of the type of surface on microbial resistance was also analysed; in statistical terms, microbial indicators were eliminated from the shelves (plastic), gurney (leather) and seats (leather) (no significant differences). This suggested that a 30 min ozone exposure was sufficient to disinfect the three types of surfaces assessed inside an ambulance. Lastly, none of the microbial indicators survived after the conventional disinfection protocol was applied (0.1% sodium hypochlorite).

Analysis of the samples collected from the interior surfaces of HNHU ambulances transporting patients with suspected SARS-CoV-2 infection showed that 5.6% of surfaces (n = 126) were positive for SARS-related coronaviruses as determined using RT-qPCR. Viral RNA was found to be within the Ct range of 30.74–38.25 for the ozonised gurney surface and the ozonised ambulance electrical panel, respectively; in ambulance 1, the Ct values were found to be 30.74 and 37.87 for the ozonised gurney (R2) and ozonised seat (R3), respectively; in ambulance 2, the Ct values were found to be 36.11 and 37.41 for the non-ozonised ambulance electrical panel (R3) and the ozonised ventilation grid (R3), respectively; and in ambulance 3, the Ct values were found to be 35.96, 32.97 and 38.28 for the ozonised ambulance electrical panel (R1), ozonised gurney (R3) and ozonised ambulance electrical panel (R1), respectively. Ct values of the internal positive quality control for SARS-related coronaviruses were 26.39 ± 0.51.

## 4. Discussion

In vitro results show that exposure to a 25 ppm concentration of ozone for 30 min is sufficient to disinfect the polystyrene surfaces inoculated with the four microorganisms, confirming that ozone efficacy depends on the exposure time [14,15].

Ozone efficacy varies based on the type of microorganism [16]. This can be seen when comparing the decimal reduction values for CA (D = 2.65 min), EC (D = 3.14 min) bacteriophages (D = 5.01 min) and SA (D = 5.40 min). According to Oner and Demirci [4], the microorganism’s membrane composition and complexity are significant factors for ozone action, as ozone damages the cell envelope, causing leakage and loss of intracellular components, thereby resulting in lysis and cell death. It also affects unsaturated lipids, proteins and enzymes embedded in the cell wall of gram-negative bacteria [15]. On the other hand, this effect is mediated by peroxidation and damage of polyunsaturated fatty acids on the cell membranes of bacteria, mould, yeast, and spores, as well as of S proteins and lipids in the capsid and enveloped viruses, indirectly damaging DNA and RNA [6].

The results of in situ ozone treatment in ambulances show that a 30-min exposure time at 25 ppm can eliminate microorganisms inoculated onto the gurney, seats and shelves, which are the closest points to a patient transported inside an ambulance. These results confirm the efficacy of ozone previously observed in in vitro tests (Figure 1). On the other hand, the virucidal efficacy of ozone does not depend on the type of surface [17], which was confirmed by the survival rates found in the tests performed in conventional ambulances (Table 1).

According to Traoré et al. [18], the survival capacity of nosocomial pathogens on ambient-exposed surfaces is inversely proportional to the air temperature, as higher temperatures accelerate the inactivation or reduction of microbial populations. When comparing the initial populations of the four microbial indicators in in vitro (22 ± 2 °C) and in situ assays in ambulances (26 ± 2 °C), a 1 log decrease was observed. Following ozone exposure, relative humidity decreased from 72% to 53%. 

Viral RNA remained within the established parameters, validating its concentration, quality and integrity [19,20]. RT-qPCR tests confirmed the presence of SARS-related viral aerosols at a rate of 5.6%, which has been reported in healthcare settings [21]. Viral particles can be transmitted through air microdroplets or direct contact and are viable at 6 °C at 50% RH for 5 days [22,23].

Ozone is known to be the third most powerful oxidising agent after fluorine and persulfate and can induce oxidative stress in living organisms. The WHO describes ozone as one of the best biocides against microbial agents. Ozone has also been reported to kill viruses, such as enterovirus, poliovirus, rhinovirus and murine coronavirus, which are included within Group IV together with SARS-CoV-2 (Baltimore classification), a group characterised by positive-sense, single-stranded RNA [6]. Some of the advantages of ozone sterilisation are that, being a gas, it can infiltrate all areas and is therefore more efficient than manually applied disinfectant liquids or sprays [24].

The evidence obtained demonstrates the advantages of using ozone as a disinfectant to reach inaccessible areas of the ambulance, ease of application, low risk of contamination to health personnel and has no residual products in the environment or on the surface compared to the use of ultraviolet light and quaternary ammonium [25].

Van Doremalen et al. [2] were uncertain whether virus transmission through contaminated surfaces or aerosols was viable for 3 h. However, virus viability depends on antibiotic conditions, such as humidity (>90% RH). Adequate disinfection with maximum antiviral efficacy cannot be achieved with a humidity <50% [6,26,27,28]. During the in-situ experiments conducted in ambulances, humidity was found to be <90% (55–75% RH), which could account for the detection of SARS-related coronaviruses. Another relevant aspect is the type of surface sampled; most surfaces with viral load are steel and metal, which is consistent with the report by Fathizadeh et al. [29]. This coincides with the findings obtained in the COVID-19 ambulances.

Molecular detection of SARS-CoV and influenza viruses does not necessarily involve the presence of viable viruses. However, amplifications can be made, as observed in this study, which detected positive SARS-related coronavirus samples. Quevedo-león et al. [25] state that following 20 ppm ozone exposure for 10 or 15 min, viruses were inactivated and showed envelope damage, resulting in single-stranded RNA damage or impaired binding to normal cells. This has also been reported by Bayarri et al. [30] for different respiratory pathogen viruses. Moreover, Cristiano [31] states that a 30 min period is needed to treat ambient surfaces contaminated with viruses.

In retrospect, concentrations higher than 0.08 ppm of ozone can induce negative health effects (headache, nausea, fainting, etc.). In this case, there was no direct exposure of the gas to health personnel because the equipment was operated remotely inside a closed and empty ambulance, and was followed by a ventilation protocol for 15 min, reducing the risk to health [25,32].

Complementary evaluations in biological samples (sputum, saliva, fluids, and blood) would be recommended to confirm the effectiveness of ozone as a disinfection method. Likewise, in recent years, the presence of aldehydes in low concentrations has been reported from the interaction of ozone with plastic materials and cloth, unlike metallic materials that do not leave residual products [33]; this should be taken into account in future investigations.

## 5. Conclusions

This study shows that exposure to 25 ppm ozone for 30 min using the OzGe prototype leads to low survival rates for microbial indicators (0–5%) on the interior surfaces of ambulances. Molecular tests allowed for the detection of SARS-related coronaviruses in seven samples (5.6%) collected from surfaces inside the ambulance.

## Figures and Tables

**Figure 1 ijerph-20-05776-f001:**
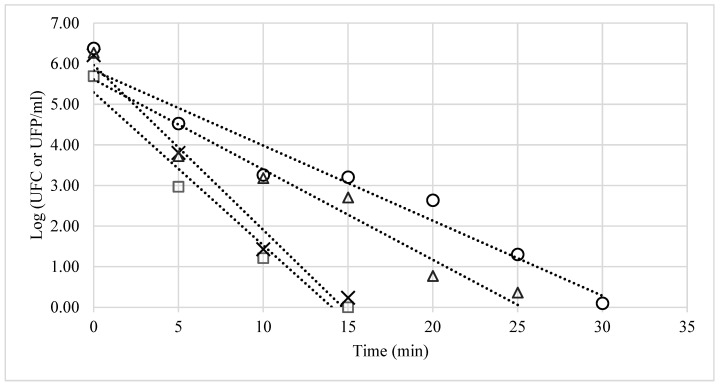
Survival curves for bacteria exposed to 25-ppm ozone for 30 min. The corresponding decimal reduction times (D) were: *Candida albicans* (□) D = 2.65, R^2^ = 0.97; *Escherichia coli* (×) D = 3.14, R^2^ = 0.92; *Salmonella phage* (△) D = 5.01, R^2^ = 0.95 and *Staphylococcus aureus* (○) D = 5.40, R^2^ = 0.93.

**Table 1 ijerph-20-05776-t001:** Survival rate of microbial indicators inoculated (SA, EC, CA and Φ21) after the exposure of seat surfaces, gurneys, and shelves inside conventional ambulances to 25 ppm ozone for 30 min (n = 3).

	Seats			Gurneys			Shelves		
	N_o_	N_f_	%SR	N_o_	N_f_	%SR	N_o_	N_f_	%SR
*SA*	5.3 ± 0.12	0.05 ± 0.16	1.0	5.7 ± 0.29	0.2 ± 0.27	3.1	6.0 ± 0.10	0.0	0.0
*EC*	5.3 ± 0.22	0.0	0.0	5.4 ± 0.30	0.0	0.0	6.3 ± 0.12	0.0	0.0
*CA*	5.0 ± 0.18	0.0	0.0	5.1 ± 0.09	0.0	0.0	5.3 ± 0.09	0.0	0.0
*Φ* *21*	5.7 ± 0.26	0.0	0.0	6.2 ± 0.25	0.0	0.0	6.5 ± 0.23	0.3 ± 0.43	5.3

N_o_: Initial population of the microbial indicator, N_f_: Final population of the microbial indicator after ozonisation, SA: *S. aureus*, EC: *E. coli*, CA: *C. albicans*, Φ21: *S. phage*.

## Data Availability

Data sharing not applicable.

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
