# Peer review of "Ozone Efficacy for the Disinfection of Ambulances Used to Transport Patients during the COVID-19 Pandemic in Peru"

_ijerph, 2023, doi:10.3390/ijerph20105776_

Round 1

Reviewer 1 Report

There are few studies that evaluate the ozone disinfection mechanism against COVID-19, specifically. Likewise, it has been shown that COVID-19persists on surfaces for up to 72 hours, while other coronaviruses have persisted for up to 9 days on surfaces. Therefore, the study represents a significant contribution to the behavior and stability of the virus. However, although ozone in its gaseous state is more reactive, disinfection using aqueous ozone is more reactive due to the ease of monitoring the concentration of oxidants. 

I believe that the authors should include the safety problems that persist when ozone aerosols are implemented, considering compliance with the guidelines on ambient air quality. Despite this, research on its efficacy for virus inactivation on surfaces is scarce.

There are several issues that need to be addressed to ensure safe and effective use of ozone in gaseous form. Therefore, it is suggested that the authors describe the formation of by-products inside ambulances from materials that can come into contact with ozone during disinfection.

The writing requires to be reviewed by a native of the language.

Author Response

We thank you for your review. We upload a response to Reviewer in the archive.

Reviewer 2 Report

Dear Authors,

I analyzed the paper, and from my review, I made the following observations:

General Comment:

The topic has already been dealt with in the literature in different backgrounds. Still, the data analyzed here enriches and confirms other scientific work, adding some new elements that will need further investigation.

This study has a sound methodology and a clear description of the steps. In addition, the study is relevant to current scientific practice. However, it might be a “niche” topic in biomedical research and not all readers can follow through the steps of the experiments.

The aim of the research is well specified in the introduction. I greatly appreciated the description of the materials and methods, although I have a few observations about some study limitations. It would certainly be appropriate to specify better the division between the analysis of the effects of ozone on microbial and the subsequent phase devoted to Sars-cov-2.

Title: I suggest adding that the research was conducted in Peru. Also, the title only specifies the transport of patients with coronavirus, whereas in reality, they represent only the minority of those transported in the study.

Abstract: The abstract is complete and clear. It explains the paper's aims well and includes all the necessary parts. There is only one mistake, Line 21, you talk about patients with COVID-19, while in the results section Line 182 you talk about “ with suspected or confirmed COVID-19”. Which one is correct?

Keyword:  I suggest to cancel “ decimal reduction”

Introduction: This part is well written, but Authors can better explain the works in the literature about this topic. Also when you talk about “ At Present” you need to give more details about the period.

The background might help readers to understand the current status of research with some expansion. Fomites contamination and disinfection has been widely investigated, and several different approaches are being researched right now. To complete this perspective, the discussion might help in this by addressing: "What is the main advantage in Ozone compared to UV technology?”, or It might be useful to compare the strengths and weaknesses of the two different approaches.

Material and Methods: This section is divided into sub-headings, which makes it easier to read. The explanation of the three phases that make up the study is clear and precise. I also appreciated that all the places where the studies were made were well explained.

Exploratory field testing in ambulances transporting patients with COVID-19: please provide some information about the type of ambulance and vehicle dimensions etc. It can be useful to compare the in vivo test to the in vitro experiment made into a 23m3 room.

Line 65: “Peru” is repeated.

Line 66: Please provide Some more description of the UV lamps used in this study, like wavelength spectrum, power in UVC etc.

Line 66 – 67. It is said, “Each of them is 150 × 60 × 60 cm in size”. Do the Authors describe the dimension of the lamp? To my knowledge, there is no lamp that is of such dimension. Do the Authors describe the part in which the lamp/s  is/are placed? Please specify.

Line 101: In how much time does the ozone concentration reach the 25-ppm threshold? It might be relevant to evaluate the practicality of this disinfection method.

Line 115: you have to explain here why you choose this places of the ambulance and not later in the text

Results:

In my opinion, it would be better if the patients transported with the ambulances were all with covid-19 confirmed.

Figure 1: There is a mistake in the CA. The 15 min at 0 is Missing in the graph. Also, the different symbols are not well clear in sight. A better formatting is necessary.

Line 166: Add Space between the Figure and Main text

From lines 170 – 174 it is said, “The effect of the type of surface on microbial resistance was also analyzed, in statistical terms, microbial indicators were eliminated from the shelves (plastic), gurney (leather) and seats (leather) (p-value> 0.05). This suggested that a 30-min ozone exposure was sufficient to disinfect the three types of surfaces assessed inside an ambulance.”. It is not clear whether the authors want to highlight significant differences or not. The 'p' value does not show a significant difference in the comparison. In fact, it says "p>0.05". Does this mean that there are no significant differences between the materials? Then "p>0.05" is correct. If, on the other hand, there are significant differences, it should read "p<0.05". From the text, it would appear that it is the latter, so please specify more clearly or correct the typo.

Table 1: Needs an overhaul in formatting, as I would suggest decreasing the size of the text and creating a different model for the table itself. There is a second horizontal line starting from N0 of Gurneys. Delete it

Table 2: it can be explained well in the text

Discussion: This section is well-written and clear. Reference n. 21 is not so clear; improve this part.

A paragraph on the limits of research should be developed.

The limitations of using ozone should also be discussed. For example, repeated use degrades tyres and can cause damage to materials. Even in an ambulance, we therefore have critical aspects that should be managed.

Reference: The number of references is good, but Increasing the number of scientific papers is recommended. There have been more works in recent years on this topic.

Author Response

We thank you for your review. We upload our response in the archive.
